# Effects of Adding Chitosan on Drug Entrapment Efficiency and Release Duration for Paclitaxel-Loaded Hydroxyapatite—Gelatin Composite Microspheres

**DOI:** 10.3390/pharmaceutics15082025

**Published:** 2023-07-26

**Authors:** Meng-Ying Wu, I-Fang Kao, Chien-Yao Fu, Shiow-Kang Yen

**Affiliations:** 1Department of Materials Science and Engineering, National Chung Hsing University, Taichung 40227, Taiwan; slimu16882013@gmail.com (M.-Y.W.);; 2Department of Orthopaedics, National Defense Medical Center, Taipei 11490, Taiwan; yao0607@yahoo.com.tw; 3Department of Orthopaedics, Taichung Armed Forces General Hospital, Taichung 40705, Taiwan

**Keywords:** hydroxyapatite—gelatin microspheres, cone-like pores, paclitaxel, drug entrapment efficiency, chitosan, release duration

## Abstract

Hydroxyapatite—gelatin microspheres with cone-like pores were synthesized via the wet-chemical method using ammonium dihydrogen phosphate ((NH_4_)H_2_PO_4_) and calcium nitrate (Ca(NO_3_)_2_·4H_2_O) as a source of calcium and phosphate ions with the addition of gelatin, which proved to be more osteoconductive than commercial products, such as fibrin glue and Osteoset^®^ Bone Graft Substitute. Following the method of the previous study for loading paclitaxel (PTX), a drug entrapment efficiency of around 58% was achieved, which is much lower than that of the doxorubicin (DOX)-loaded one. Since PTX is hydrophobic while DOX is hydrophilic, the order of chitosan processing and addition of the solvent were tuned in this study, finally leading to an increase in drug entrapment efficiency of 94%. Additionally, the release duration of PTX exceeded six months. The MTT assay indicated that the effect of drug release on the suppression of cancer cells reached more than 40% after one week, thereby showcasing PTX’s capacity to carry out its medicinal functions without being affected by the loading procedures.

## 1. Introduction

Hydroxyapatite (HAp) is a naturally occurring mineral that is widely utilized in a variety of fields, including biomaterials, medicine, and tissue engineering. It is the main component of hard tissues in the human body, such as bones and teeth, and it provides strength and stability to these tissues [1]. It has many properties that make it an excellent biomaterial. For example, it is biocompatible, meaning that it has no immunological response or other negative effects on the body. This property is essential for any material that will be implanted or used in medical applications. HAp is also non-toxic and non-inflammatory, which further enhances its biocompatibility [2]. Another important property of HAp is its ability to integrate with bone tissue. When HAp is implanted into the body, it can bond with the surrounding bone tissue and become a part of the body’s natural structure. This property makes HAp an excellent material for bone repair and regeneration [3,4,5]. HAp has many applications in medicine and tissue engineering. For example, it is commonly used in dental implants and bone grafts to promote bone growth and integration. It is also used in cardiovascular stents to promote healing and prevent restenosis [6]. In addition to its use as a biomaterial, HAp is also used as a carrier for drugs, DNA, and proteins. HAp particles can be loaded with these substances and delivered to specific areas of the body, where they can be released over time. This approach is particularly useful for targeted drug delivery and gene therapy [7].

In a previous study we conducted, it was shown that by employing a wet-chemical technique, composite microspheres of HAp that are porous in nature and contain biodegradable gelatin can be effectively manufactured [8]. There is a study that evaluated the ability of HAp–gelatin (HAp-Gel) composite scaffolds to facilitate the growth of bone tissue in a rat calvarial lesion model, and they demonstrated the highest potential for osseointegration, leading to 90% new bone formation in the final stage [8]. In addition, there was also a report that used electrochemical deposition to coat doxorubicin–chitosan–HAp composite materials onto titanium alloys for cancer treatment [9].

Cancer is characterized by the uncontrollable growth of abnormal cells that ultimately form malignant tumors, and the invasion of nearby tissues and distant organs by these tumor cells is the main contributor to both illness and mortality in most cancer patients [10,11]. Cancer has emerged as a significant public health concern in this century due to its elevated occurrence and fatality rates [12]. Multiple approaches can be employed for the treatment of cancer, encompassing surgical procedures, radiation therapy, chemotherapy, and targeted therapy [13]. Among these, chemotherapy is a commonly used treatment that refers to the administration of anti-cancer medications to either impede the growth of malignant cells or cause direct harm to them. Although chemotherapy is primarily recognized as a systemic treatment for cancer, it is also possible to administer it using non-systemic approaches [14]. Nonetheless, the treatment process can have an impact on the entire body, not only specifically targeting cancerous cells but also affecting healthy ones. As a result, chemotherapy can cause side effects such as oral ulcers, vomiting, and hair loss [15,16]. To surpass the restrictions of chemotherapy, various systems for delivering drugs have been developed.

Conventional drug delivery systems do have some limitations, such as nonspecific delivery, poor solubility, low absorption resulting in low bioavailability, and suboptimal drug concentrations in the bloodstream, which can lead to side effects when drugs accumulate in other organs. Modulating the delivery of existing drugs represents a straightforward and practical alternative to developing new medications [17].

Targeted drug delivery systems offer significant potential for treating life-threatening diseases, especially cancer and its related malignancies. It is a method of drug delivery that delivers the drug directly to the patient and increases the concentration in a specific area, with the advantage of requiring only a small dose to reach the ideal effect, thus reducing the side effects of the drug. The core of targeted drug delivery systems is the carrier; common carriers include hydroxyapatite, liposomes, nanoparticles, and polymeric micelles [18,19,20]. These carriers have different characteristics, advantages, and disadvantages, and their drug delivery features can be tuned by modifying their physicochemical properties. For example, the shape and porosity of HAp can affect the loading and release of drugs. The smaller the porosity, the more significant the initial burst release of the drug will be [6].

The shape and porosity of HAp particles should be carefully controlled to optimize drug loading and release and minimize any adverse reactions. This can be achieved by using different manufacturing techniques, such as varying the temperature, pH, and stirring rate during synthesis [21,22,23]. By carefully controlling the shape and porosity of HAp particles, it is possible to further optimize their performance as drug carriers and minimize any potential adverse effects.

Chitosan was first discovered and discussed in 1859 [24]. It is made by deacetylating chitin, which is present in the exoskeletons of crustaceans like crabs and shrimp [25]. Its cationic features are represented by the amino group at the C-2 position of the glucosamine unit, which also confers innate qualities including wound healing, antibacterial activity, and adhesiveness, making it a suitable carrier for drug delivery systems. Chitosan can also be utilized in different forms depending on the functionality and application of the carrier, such as controlled drug delivery, transfection, in situ gelation, penetration enhancement, colonic targeting, and inhibition of efflux pumps. Its properties, such as biological adhesiveness, also allow it to adhere to soft and hard tissues, making it useful in many applications in orthopedic, dental, ophthalmic, and surgical measures. In nasal delivery, chitosan nanoparticles or liposomes loaded with Streptococcus equi extracts can induce both Th1 and Th2 responses. Additionally, chitosan nanoparticles can facilitate antigen uptake by antigen-presenting cells and prolong the antigen’s residence time in the nasal cavity [26].

Chitosan also shows behaviors such as biocompatibility, biodegradability, hydrophilicity, and non-toxicity, and thus it is widely used in environmental and biomedical engineering. In addition, some clinical studies have explored the potential uses of chitosan, including its application as a scaffold material for tissue engineering as well as nerve regeneration tubes and cartilage regeneration. It has also been studied as an adjuvant in surgery to promote wound dressing [27,28,29]. A report has been published that combines the advantageous characteristics of chitosan and polyhedral oligomeric silsesquioxanes nanoparticles in order to develop novel nanoparticles for the purpose of bone tissue regeneration [30].

The cell cycle consists of four stages. The G-1 phase is the period when the cell prepares for division. During the S phase, DNA synthesis occurs to replicate genetic material. During the G2 phase, there are metabolic alterations that enable the assembly of the materials required for the processes of cell division and mitosis. The final phase is the M phase, which is characterized by cell division through the processes of mitosis and cytokinesis [31]. The cell cycle is a fundamental process that is crucial for the proper functioning of all living organisms. Any disruptions or errors in the cell cycle can have serious consequences, including the development of genetic mutations, cell death, or cancer. 

Pacific yew tree bark is the source of paclitaxel (PTX), a natural plant alkaloid and diterpenoid compound. It exhibits the characteristics of a white crystalline powder, possessing a molecular formula of C_47_H_51_NO_14_ and a molecular weight of 853.9 g/mol [10,32,33,34]. PTX is an anti-cancer medication that is designed to target microtubules, an essential element in the formation of spindle fibers during the process of cellular division. Microtubules also play an essential role in maintaining cell structure, motility, and intracellular movement. The assembly of microtubules takes place during the G2 and pre-phase of mitosis. The chemical composition of the substance comprises a taxane ring with an attached oxetane side ring and a homochiral ester side chain [35]. PTX stabilizes microtubules, inhibiting their depolymerization and thus preventing cell division. In consequence, the cell cycle stops at G2/M phase and eventually undergoes apoptosis, preventing cell replication and growth [33,34,35,36]. PTX is frequently employed to treat several types of breast, ovarian, lung, melanoma, and esophageal cancers [34,37].

Although PTX cannot penetrate the blood–brain barrier when administered intravenously [10,38], it has better drug transport characteristics compared to other chemotherapy drugs when administered locally. The hydrophobicity of PTX is an obstacle for systemic administration, but it is an advantageous characteristic for interstitial drug delivery since it enables accurate release during polymer degradation [36]. Research has indicated that administering PTX locally, such as through intra-tumoral injection, can result in elevated drug concentrations within the tumor tissue, thereby minimizing systemic side effects. This localized delivery approach can improve the efficacy of PTX in treating various types of cancer [39,40,41]. Moreover, recent advancements in nanotechnology have allowed for the development of PTX-based nanoformulations that can further enhance the drug’s localized delivery and efficacy. For instance, PTX can be encapsulated in nanoparticles, liposomes, or micelles, which can enhance the drug’s solubility and targeting potential while shielding it from degradation. Preclinical and clinical studies have demonstrated that PTX-based nanoformulations have exhibited promising outcomes, indicating their potential for improving cancer treatment [38,42,43].

As with doxorubicin (DOX), which has been loaded on porous HAp-Gel microspheres in previous research [44], a similar method is utilized to load PTX on porous HAp-Gel microspheres. However, DOX is hydrophilic, while PTX is hydrophobic. Commercial formulations usually dissolve PTX in castor oil, which can cause serious toxic reactions such as hypersensitivity, hyperlipidemia, abnormal lipoprotein morphology, red blood cell aggregation, peripheral neuropathy, bone marrow suppression, and arrhythmia [11,45,46,47]. Therefore, in this experiment, alcohol was used as the solvent instead. However, the solubility of chitosan is much lower in alcohol than in aqueous solutions for hydrophilic DOX [48]. Therefore, the preparation process and method may need to be adjusted. The primary aim of this research was to dissolve PTX in alcohol to form a nanocell with outer hydrophilicity and inner lipophilicity. This way, the nanocell can form a hydrogen bond with porous hydroxyapatite and/or chitosan through its hydrophilic nature. The aim was to achieve a drug entrapment efficiency of more than 90% and a drug release duration of more than six months, as well as inhibitory effects on cancer cell proliferation.

## 2. Materials and Methods

### 2.1. Preparation of Aqueous Solutions and Precipitation of HAp-Gel

Aqueous solutions of 0.025 M ammonium dihydrogen phosphate ((NH_4_)H_2_PO_4_), SHOWA, Japan), 0.042 M calcium nitrate (Ca(NO_3_)_2_∙4H_2_O, SHOWA, Tokyo, Japan), and 0.111 M gelatin (Fluka Chemical, Biochemica 48,723 (bloom 160), Buchs, Germany) were meticulously prepared and placed in a heating water bath to maintain the chemical solution at 65 °C with constant stirring at 120 rpm for 30 min. Three solutions were then mixed, and the HAp-Gel was allowed to precipitate at 65 °C. The precipitate was then separated from the supernatant using vacuum filtration and subsequently washed multiple times with ultrapure deionized water to eliminate residual gelatin. If gelatin is not washed properly, the final sample will not show any porosity on HAp-Gel. The washed precipitate was then dried at 45 °C for at least 48 h using a high-precision oven to maintain sample integrity and collected for further experimentation. 

### 2.2. Fourier Transform Infrared Spectroscopy Analysis (FTIR)

Commercial HA (Sigma-Aldrich, St. Louis, MO, USA) and HAp-Gel powders were mixed with KBr at a ratio of 1:100 for Fourier transform infrared spectroscopy (FTIR) analysis, using a wave number range of 4000 to 400 cm^−1^ to characterize chemical bonds.

### 2.3. Loading of Drugs Onto HAp-Gel and/or Chitosan

#### 2.3.1. Creating Calibration Curve for PTX Concentration Analysis

Due to the hydrophobicity of PTX (Sigma-Aldrich, St. Louis, MO, USA), we dissolved it in an alcohol solution with a concentration range of 25 to 150 ppm, measured its absorbance using UV-Visible spectroscopy (Hitachi U-3010, Tokyo, Japan), and used the resulting absorbance values to create a calibration curve that adhered to established standards. The resulting calibration curve was then used for the quantitative analysis of unknown PTX in samples based on their absorbance values. The related chemical structure of PTX, shown in Figure 1a, can be dissolved in alcohol to form a nanocell with outer hydrophilicity and inner lipophilicity.

#### 2.3.2. Drug Loading Process for PTX Divided into Two Categories

A quantity of 1.5 mg of PTX was dissolved in a 75% ethanol solution. Chitosan (from crab shells, Sigma-Aldrich Co., USA) solution, dissolved in 0.33 vol% acetic acid, was added or omitted to achieve PTX–ethanol–chitosan solutions containing 0% or 0.125% chitosan, respectively; then, 20 mg of HAp-Gel powders was added in and shaken for 48 h, and the final samples are referred to as P0 and P1. As the drug loading time increases, the entrapment efficiency also increases. However, after 40 h, the entrapment efficiency reaches a saturation point. Therefore, we fixed the entrapment time at 48 h. Upon completion of the designated loading duration, the samples were vacuum-freeze-dried at −80 °C for 24 h to remove moisture, and the drug-loaded HAp-Gel powder was carefully collected and stored at a low temperature of −20 °C. The chemical structures of chitosan and HAp are shown in Figure 1b,c. They are full of hydroxyl and/or amino function groups which can combine each other as well as the PTX–alcohol nanocell via hydrogen bonds.

The second approach involved placing 20 mg of HAp-Gel in a 75% ethanol solution containing 1.5 mg of PTX. The mixture was shaken for 24 h. After freeze drying at −80 °C for 24 h, a 0.125% chitosan solution was added to the dried mixture, and the shaking process was continued for an additional 24 h. Following this procedure, the drug-loaded sample, identified as P2, underwent the same freeze drying and collection steps mentioned before.

#### 2.3.3. Bonding Capabilities of Chitosan with PTX and HAp-Gel

A total of 1.5 mg of PTX was dissolved in 75% ethanol, followed by the addition of a 0.125% chitosan solution. The resulting mixture was shaken for 48 h and subsequently underwent freeze drying. This specific sample is assigned to C1.

Chitosan was mixed with 20 mg of HAp-Gel for 2 h and 24 h; subsequently, a solution containing 1.5 mg of PTX was added in, and the mixture was shaken for another 48 h. The final samples are referred to as C2 and C3. Afterward, the mixture underwent vacuum freeze drying at −80 °C for 24 h. Samples C1, C2, and C3 were prepared for the investigation of the bonding capability of chitosan with PTX and HAp-Gel according to drug entrapment efficiency (DEE).

Samples P0, P1, P2, C1, C2, and C3 were all shaken at a speed of 80 rpm. The sample was removed after the freeze-drying process and the residual PTX in the original test tube was dissolved in ethanol. The absorbance of the supernatant was measured using UV-visible spectroscopy. The obtained absorbance values were then used to determine the concentration by referring to the calibration curve. The DEE and drug loading content (DLC) were calculated using the following formulas [41]:
(1)$$\begin{array}{c} \text{DEE}\;\left(\%\right)\;=\\ \left(\text{Initial}\;\text{PTX}\;\text{amount}\;-\;\text{PTX}\;\text{amount}\;\text{in}\;\text{the}\;\text{supernatant}\right)\;\times \;100/\text{initial}\;\text{PTX}\;\text{amount}, \end{array}$$
(2)$$\begin{array}{c} \text{DLC}\;\left(\%\right)\;=\\ \left(\text{Initial}\;\text{PTX}\;\text{amount}\;-\;\text{PTX}\;\text{amount}\;\text{in}\;\text{the}\;\text{supernatant}\right)\;\times \;100/\text{carrier}\;\text{amount}, \end{array}$$


### 2.4. Surface Morphology of HAp-Gel, P0, P1, P2, C1, C2, and C3

The dried samples, including HAp-Gel, P0, P1, P2, C1, C2, and C3, were mounted onto a sample holder using carbon tape for lower magnification, while higher magnification was used for the Au-coated sample, and the samples were then subjected to analysis using scanning electron microscopy (SEM, JSM-5400) to evaluate their surface morphology. The microscope was operated at a working distance of 12 mm, a probe current of 30 mA, and a range of 10–20 kV.

### 2.5. Drug Release from PTX-HAp-Gel

To conduct the release experiment, 15 mL centrifuge tubes were used, and 10 mg samples were placed inside them along with 10 mL of phosphate-buffered saline (PBS, pH 7.4, Gibco, Billings, MT, USA) solution. The experiment was carried out in a water bath maintained at 37 °C with a shaking speed of 80 rpm. At designated time intervals, the sample and solution were subjected to centrifugation, and 1 mL of the supernatant was collected for measuring the absorbance of PTX using UV-visible spectroscopy. The frequency of measurements was as follows: every two hours on the first day until the seventh hour; once daily during the following week; every four days over the next three weeks; every two weeks for the following two months; and once a month after the two-month period. After collecting 1 mL of the supernatant, the same volume of PBS was added to the original tube. The obtained values were then substituted into the calibration curve to calculate the released drug concentration. Each measurement result was subsequently used in the following formula to generate the cumulative drug release profile [41]:(3)$$Mc=Mt+\frac{v}{V}\sum_{0}^{t-1}Mt$$

In this equation, Mc represents the corrected mass at time t, Mt denotes the apparent mass at time t, v signifies the volume of the sample taken, and V represents the total volume of the release fluids.

### 2.6. Cell Experiments

#### 2.6.1. Cultivation of Human Osteosarcoma Cells (G292, ATCC CRL-1423)

To establish a suitable culture environment for the cells, McCoy’s 5A (Modified, Gibco, Waltham, MA, USA) medium was selected as the culture medium. To minimize the risk of bacterial contamination, penicillin–streptomycin was added to the medium. Additionally, to promote cell proliferation, 10% fetal bovine serum (Biological Industries, Beit Haemek, Israel) was also added to the medium, which provides growth factors essential for cell growth [49]. The cells were then cultured under standard conditions in a CO_2_ incubator maintained at 37 °C and 5% CO_2_.

#### 2.6.2. Direct Cell Toxicity Test

PTX-HAp-Gel samples with different chitosan concentrations, including P0, P1, and P2, were utilized for toxicity testing. Pure PTX served as the positive control, while cells without any treatment were used as the negative control. The samples and pure PTX powder, containing the same amount of PTX, 2.5 µg, were placed in a 24-well plate and treated with UV light for sterilization. Subsequently, 10^4^ cells/100 µL were seeded into each well containing the samples and co-cultured at 37 °C with 5% CO_2_. On days 1, 4, 7, and 14, cell viability was evaluated using the MTT (3-(4,5-dimethylthiazol-2-yl)-2,5-diphenyltetrazolium bromide) assay. To maintain cell growth, half of the culture medium was replaced every three days.

#### 2.6.3. MTT Assay

To evaluate cell activity, an MTT assay was performed. Firstly, the MTT (M5655, Sigma-Aldrich, USA) stock solution was diluted using McCoy’s 5A medium to create a working solution with a concentration of 0.5 mg/mL. At specific time points, the culture medium was aspirated from the wells, and each well was supplemented with 0.5 mL of the MTT working solution. The plate was then placed in an incubator for three hours to facilitate the formation of formazan crystals. After the incubation period, the MTT solution was extracted, and an equivalent volume of dimethyl sulfoxide (Merck, Darmstadt, Germany) was added to each well to dissolve the formazan crystals produced by the live cells, resulting in a purple liquid [50]. The absorbance of this liquid at 545 nm was measured using an ELISA reader (Stat Fax-2100, Awareness Technology, Inc., Palm City, FL, USA) to determine the cell activity.

### 2.7. Statistically Analysis

The Student’s *t*-test in the Microsoft Excel statistical program was used to compare the variations in DEE, DLC, and cell viability between the groups during the experiment. It was deemed statistically significant when the *p*-value for the difference between the groups is less than 0.05.

## 3. Results and Discussion

### 3.1. Fourier Transform Infrared Spectrograph Spectrometer

The FTIR spectra of the commercial HA, HAp-Gel composite microspheres, and biopolymer are shown in Figure 2. In the FTIR analysis, mainly the peaks for PO_4_^3−^ and OH^−^ groups in the commercial HA and HAp-Gel microspheres can be identified. The 470 cm^−1^ adsorption peak is derived from the ν_2_ PO_4_^3−^ degenerate bending mode. The peaks at 558 and 605 cm^−1^ are the ν4 bending vibration modes of PO_4_^3−^, and the peak at 963 cm^−1^ is assigned to ν_1_, the symmetric P-O stretching bond. The strong one at 1033 cm^−1^ resulted from the ν_3_ P-O asymmetric stretching of PO_4_^3−^ [51,52]. The broad band around 3100–3400 cm^−1^ corresponds to the adsorbed hydrate, and the sharp peak at 3571 cm^−1^ is the stretching mode of the OH^−^ group. In addition, absorption peaks at 865 and 1456 cm^−1^ indicate the CO_3_^2−^ group, which is the result of the absorption of CO_2_ in air dissolved in solution, meaning that the carbonated substitution took place. The pure biopolymer shows typical amide bands at 1656 and 1544 cm^−1^. The amide I band at 1656 cm^−1^ is attributed to C=O stretching. The amide II band at 1544 cm^−1^ is due to the coupling of the bending modes of N–H and C–N bonds. Generally, the amide I band is strong, the amide II band is moderate, and the amide III band is weak [52,53]. In the composite microspheres, except for functional groups possessed by HA, amide bands also appeared at 1656 (C=O stretching vibration) and 1544 cm^−1^ (N-H stretching vibration) but did not exist in commercial HA, indicating that the microspheres were indeed composed of hydroxyapatite and gelatin.

### 3.2. Surface Morphology of HAp-Gel

As shown in Figure 3, the HAp-Gel synthesized through the wet-chemical method assumes a spherical configuration measuring approximately 50 µm with a surface characterized by conspicuous protrusions and porous features, revealing cone-like micropores.

In toxicity assessments conducted on BEAS-2B cells, needle-like HA and plate-like HA demonstrate heightened toxicity compared to spherical and rod-shaped HA [21]. Furthermore, investigations have revealed that varying the shape and size of HA results in distinctive outcomes. Needle-like HA particles averaging 5 µm in size elicit the most pronounced inflammatory response, followed by smooth, spherical HA particles of 0.1 µm. Both rough-surfaced and smooth-surfaced HA particles of approximately 20 µm induce inflammatory reactions, whereas HA particles measuring 100 µm in size with a smooth surface do not provoke an inflammatory response [2]. The previous research conducted in our laboratory corroborates the favorable biocompatibility of HA synthesized using this method when co-cultured with osteoblast-like cells [3].

Although the XRD patterns of the prepared composite microspheres showed crystal planes of HAp, the calcium-to-phosphorus ratio of the HAp-Gel is 1.52, as depicted in the previous report [44]. Therefore, it is classified as calcium-deficient compared to the typical calcium-to-phosphorus ratio of 1.67 for HAp [54].

**Figure 3 pharmaceutics-15-02025-f003:**
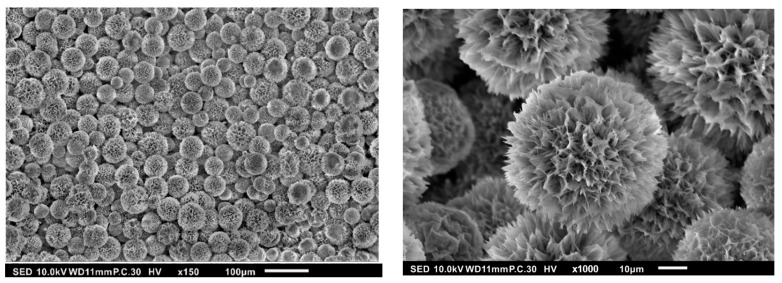
SEM images of HAp-Gel.

### 3.3. Loading of Drugs onto HAp-Gel

#### 3.3.1. Creating Calibration Curve for PTX Concentration Analysis

Figure 4a indicates the UV-visible spectrum of PTX. The PTX content is attributed to the absorbance at 227 nm in the UV-visible spectrum. For the measurements of PTX dissolved in ethanol solutions ranging from 25 to 150 ppm, a calibration curve was generated using the obtained values, as depicted in Figure 4b. The equation of the calibration curve is y = 0.0214x + 0.3063, with an R-squared value of 0. The absorbance values obtained from the subsequent measurements of drug loading and drug release were all substituted into this equation to calculate the actual drug quantity.

#### 3.3.2. Drug Loading Process for PTX

In the drug loading experiment, slight modifications were made to the drug loading process to investigate potential differences. The left diagram in Figure 5a represents the binding of PTX with ethanol, where PTX dissolved in ethanol forms microcells with an outer hydrophilic structure and an inner hydrophobic one. The former characteristics can form hydrogen bonds with chitosan and/or HAp-Gel. The right one in Figure 5a depicts the dissolution of chitosan in an acetic acid solution. The detailed procedures for each drug loading method are illustrated in Figure 5b–e for deriving samples P0, P1, P2, C1, C2, and C3. As listed in Table 1, the DEE for both P0 and P1 was around 58%, with no apparent difference observed between them. However, in the case of P2, where only the sequence of chitosan addition was tuned differently from P1, the DEE reached 94.21%.

The binding ability of chitosan and PTX in sample C1 and chitosan and HA was identified by the DEE results. The mixing durations for chitosan and HA were divided into two intervals, 2 h and 24 h, and then PTX was directly added and mixed with both components for C2 and C3, respectively. The DEE was 41.94% for C1, 58.71% for C2, and 27.57% for C3, as shown in Table 1.

Based on the above results, it can be deduced that when chitosan is mixed with HAp-Gel first, the longer the mixing duration is, the greater the bonding between chitosan and the hydroxy groups of HAp is, since more chitosan is inserted in the cone-like pore of HAp-Gel, and the surface area of chitosan and HAp-Gel, which can bind PTX, is further reduced, resulting in a lower DEE for C3. Therefore, instead of carrying out the first step for P0, achieving a DEE of 58.91%, and the second step, which is similar to that for C1, the remaining PTX can be trapped in the solution to derive sample P2, which achieves the highest DEE (94.21%), as listed in Table 1.

**Table 1 pharmaceutics-15-02025-t001:** Drug entrapment efficiencies (DEEs, %) and drug loading contents (DLCs, %) of P0, P1, P2, C1, C2, and C3.

Samples	DEE (%)	DLC (%)
P0	58.91 ± 0.14	3.53 ± 0.01
P1	57.27 ± 1.83	3.44 ± 0.1
P2	94.21 ± 0.08 *	5.65 ± 0.01 *
C1	41.94 ± 0.67 *	251.64 ± 4.07 *
C2	58.71 ± 0.88	3.52 ± 0.05
C3	27.57 ± 0.6 *	1.65 ± 0.04 *

* Statistically significant difference between P0 and P1, P2, C1, C2, and C3 groups at *p* < 0.05.

**Figure 5 pharmaceutics-15-02025-f005:**
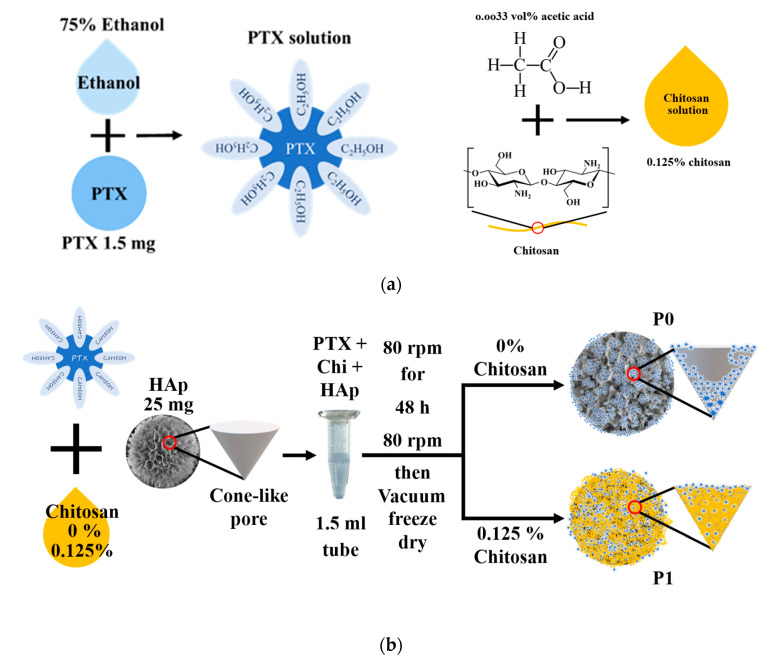
(**a**) The left diagram illustrates the dissolution of PTX in ethanol, while the right diagram depicts the dissolution of chitosan in a 0.0033 vol% solution of acetic acid. The detailed procedures for deriving samples (**b**) P0 and P1, (**c**) P2, (**d**) C1, and (**e**) C2 and C3.

The SEM images of P0, P1, P2, C1, C2, and C3 are shown in Figure 6a–f, which basically consist of the simulated model shown in Figure 5. The cone-like pores could be clearly observed only on sample P0. They totally disappeared on samples P1 and P2, and some chitosan flakes were present on their surfaces. More chitosan flakes were seen for sample C1. Fewer, smaller, and shallower pores compared with P0 were also found on samples C2 and C3, indicating that the additional 48 h of stirring led to chitosan molecular inserts in the cone-like pores, as seen for samples P1 and P2.

As the concentration of ethanol increases, the solubility of the chitosan in it decreases accordingly [45]. Due to this solubility limitation, even though the DLC of C1 is high, reaching up to 252%, as listed in Table 1, much less chitosan can be utilized as a carrier to encapsulate PTX. Furthermore, HAp-Gel also offers a shielding effect to prevent the rapid degradation of chitosan. In subsequent release experiments, it can be observed that the drug release duration is extended as a result.

**Figure 6 pharmaceutics-15-02025-f006:**
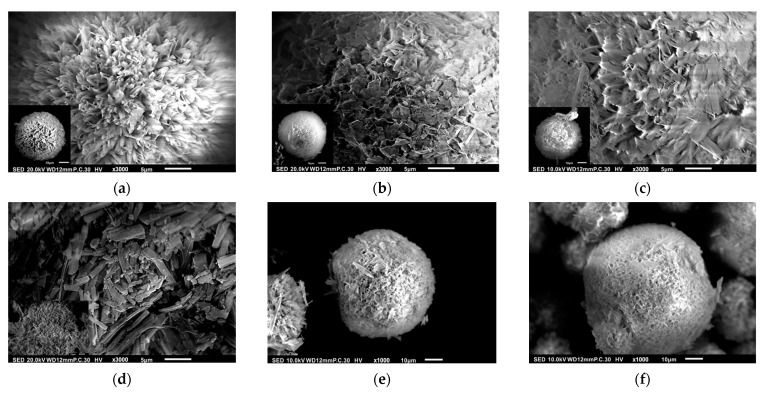
SEM observations of samples (**a**) P0, (**b**) P1, (**c**) P2, (**d**) C1, (**e**) C2, and (**f**) C3.

### 3.4. Drug Release from PTX-HAp-Gel

In the drug release experiment, samples P0, P1, and P2 were selected as the test samples. After 25 weeks, the cumulative drug release of P0 reached 94.74%, while that of P1 reached 69.14%, as shown in Figure 7a. Enlarged profiles of P0 and P1 for the initial 100 h can be divided into three stages, as shown in Figure 7b. The first stage (0–6 h) represents the initial burst phase, where residual PTX not loaded onto HAp-Gel and/or chitosan during the drying process was released. The second stage (7–30 h) corresponds to the reabsorption phase, during which P0 reabsorbed 13% of the drug and P1 reabsorbed 20%. This reabsorption is attributed to the PTX that was burst-released in the previous stage, subsequently re-binding with HAp-Gel and/or chitosan. The third stage (from 31 h onward) represents the sustaining release phase. Following the reabsorption phase, both P0 and P1 exhibited a stable and sustaining release profile, resulting from the dissolution of PTX with or without chitosan capsuled in cone-like pores, as also indicated in Figure 8a,b for the detailed processes.

Compared with a previous study we conducted [44], both studies used the same carrier, but it carried different drugs—one was hydrophilic doxorubicin, and the other was hydrophobic paclitaxel, requiring different solvents. In the drug release experiments shown in [44] Figure 7, the release of DOX-HAp-Gel without chitosan was completed within six hours, while adding 0.125% chitosan led to a release of nearly 80% within one month, with an initial burst reaching 90%. On the other hand, paclitaxel is hydrophobic, so regardless of the presence of chitosan, its release duration can last up to six months, and the initial burst is lower than that of doxorubicin release. Additionally, the release profiles of DOX-Chi and PTX-Chi also demonstrate that hydrophilic doxorubicin was released within four hours, while the hydrophobic paclitaxel took at least one month for release. Drug release in stages 1 and 2 mainly occurs via interfacial control. Nevertheless, during stage 3, the release of the drug exhibited a nearly linear trend, with slopes of P0 and P1 1.11 and 1.46, respectively. The release mechanism of the substance differs somewhat from that of PLGA or PLA, where it encompasses both the diffusion process and the hydrolysis of the polymer coating [55]. In this study, it was observed that the drug exhibited hydrophobic properties and the cone-like structure of HAp served as a protective barrier, thereby increasing the degradation of chitosan and the release of PTX.

In the release experiment of P2, the cumulative release reached 33.81% after 25 weeks, as shown in Figure 7c. Since P2 also exhibited an initial burst release in the first 10 h, its profile was enlarged and a separate release experiment was conducted for C1, as shown in Figure 7d. The release curves of P2 and C1, which can be divided into two stages, were found to be similar. The first stage (0–7 h) represents the initial burst phase of drug release from P2 involved in the release of PTX-Chi, which was not bound to HAp after drying, into the solution. Afterward, PTX detached from HAp and/or chitosan for the second stage, revealing a more durable sustaining release; related details are also depicted in Figure 8c. On the other hand, the drug-releasing rate of C1 was much greater than that of P2 due to the lack of the shielding effect provided by the cone-like pores.

P0, P1, and P2 exhibited sustained release for over six months. The pH value of PBS after six months was around 7.21 ± 0.05, showing a slight difference from the initial pH of 7.4. One possible reason for this is that upon contact of PTX–ethanol with PBS, ethanol dissolves into the solution, exposing PTX molecules. However, due to the hydrophobic nature of PTX, it does not rapidly release into PBS. Additionally, PTX–ethanol microcells filling the cone-like pores accumulated at the bottom of the HAp-Gel pores, forming larger particles, as shown in Figure 8. The larger particles of PTX–ethanol coming from the aggregation at the bottom of cone-like pores were less soluble compared to the smaller ones on the wall or pore, contributing to a slower dissolution rate. Therefore, in comparison to the hydrophilic DOX released within six hours without the addition of chitosan [41], PTX exhibited a longer release duration for P0. However, it shows some disadvantages, which indicates it is not suitable for treatments with a duration of less than six months. Nevertheless, if a more rapid rate of release is desired, it is possible to accomplish this by modifying the concentration of chitosan to increase the amount of drug, such as P0, released. The surface morphology after the drug release, observed using an SEM, for P1 and P2 was similar to that for the DOX-loaded ones [44].

**Figure 7 pharmaceutics-15-02025-f007:**
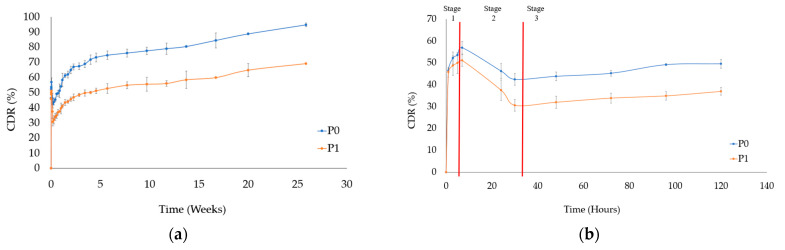
(**a**) Cumulative drug release curves of samples P0 and P1 after 25 weeks, (**b**) enlarged profiles of P0 and P1 after 1 week, (**c**) cumulative drug release curve of P2 after 25 weeks, and (**d**) enlarged profiles of cumulative drug release of P2 and C1 after 5 weeks.

**Figure 8 pharmaceutics-15-02025-f008:**
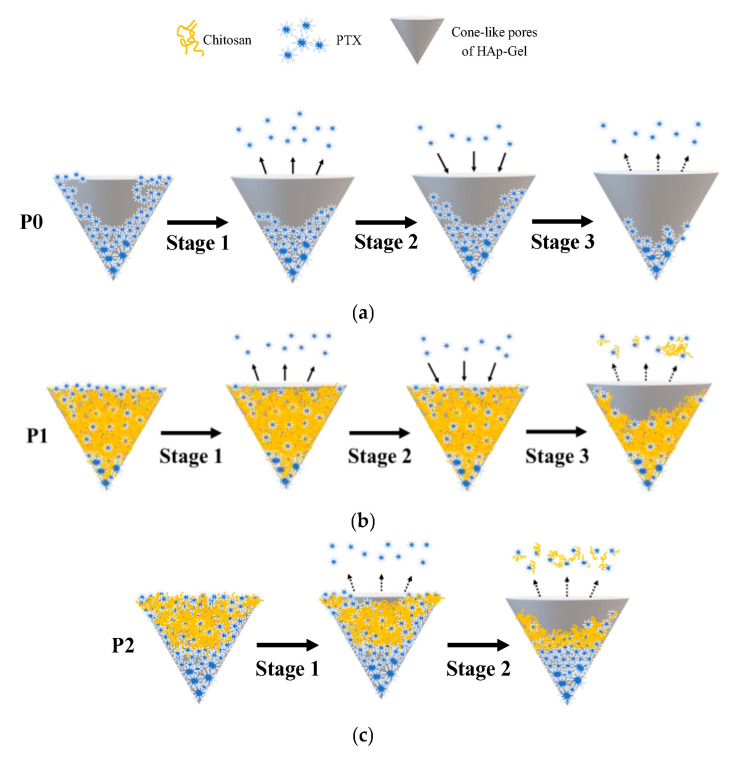
Illustrations of drug release from surface and cone-like pores of samples (**a**) P0, (**b**) P1, and (**c**) P2.

### 3.5. Cell Experiment

#### 3.5.1. MTT Assay

In the cytotoxicity test, G292 cells were co-cultured with P0, P1, P2, and pure PTX for 1, 4, 7, and 14 days. The cell viability was evaluated as shown in Figure 9. The P0 group exhibited noticeable inhibiting effects on the first day, due to the initial burst of PTX. After the fourth day, the cell viability of the P0, P1, and PTX-only groups decreased to below 50%. By the fourteenth day, except for the P2 group, the cell viability in all other groups was below 20%. P2’s cell viability remained at 56.06% on the fourteenth day, which was caused by the slower drug release of P2 (15%) than P0 (50%) and P1 (35%). However, the P2 sample still revealed a continuing trend of decreasing cell viability compared to the others. This experiment also suggests that the drug-loading methods for preparing samples P0, P1, and P2 do not affect the efficacy of PTX. An in vivo experiment using 4-week-old immunodeficient nude mice (CAnN.Cg-Foxn1nu/CrlNarl) and the G292 cell line is currently ongoing. The preliminary results indicate that PTX-HAp-Gel has an inhibitory effect.

#### 3.5.2. Observation of G292 Cell Morphology in the Cell Viability Assay

Figure 10 shows the morphological observations of co-cultured cells, with red arrows indicating HAp-Gel, blue arrows representing viable cells, and yellow arrows indicating apoptotic cells. G292 cells are adherent cells that attach and grow on the culture vessel. In the cell control group, spindle-shaped G292 cells were observed on the first day, indicating that the G292 cells had attached to the vessel and were growing normally. By the fourteenth day, the cells had proliferated and densely covered the vessel.

During the early stages of apoptosis, morphological alterations become apparent, including cell shrinkage and pyknosis. Cell shrinkage is characterized by a reduction in cellular dimensions, increased cytoplasmic density, and enhanced organelle compaction. The apoptotic cell assumes a rounded or oval morphology. Apoptotic bodies exhibit densely packed organelles within the cytoplasm, with or without accompanying nuclear fragments [56].

Through the observation of cell samples from the P0, P1, P2, and PTX-only groups, early apoptotic events were discernible as early as the first day. By the seventh day, cells exhibited a substantial apoptotic state, aligning with the findings obtained from the MTT assay.

**Figure 10 pharmaceutics-15-02025-f010:**
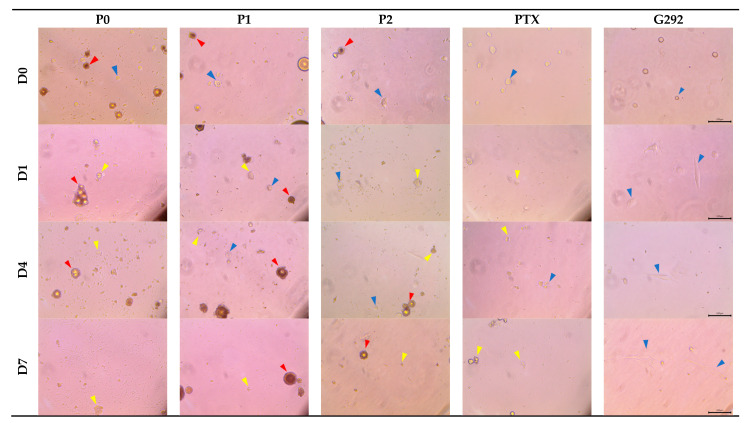
Cell morphology of G292 observed under light microscopy and photos taken at 100× magnification of samples P0, P1, P2, PTX-only, and the control group after culturing for 0, 1, 4, and 7 days. The red arrows denote HAp-Gel microspheres, the blue arrows represent viable cells, and the yellow arrows indicate apoptotic cells.

## 4. Conclusions

The hydroxyapatite microspheres (HAps) synthesized using the wet-chemical method exhibited a spherical shape with a size of approximately 50 µm. Previous studies have utilized these microspheres as drug carriers for doxorubicin. In this study, paclitaxel (PTX), another drug commonly used in cancer treatment, was selected for loading onto the microspheres. However, when PTX was loaded using a similar method to that used in previous experiments by mixing the HAp-Gel, drug, and chitosan at the same time and replacing the aqueous solvent with an alcoholic one, the drug loading efficiency was not satisfactory. Whether chitosan was absent (P0) or present (P1), the drug loading efficiency was only around 58% since PTX is hydrophobic and DOX is hydrophilic. Therefore, the order of chitosan addition was adjusted to derive P2 by adding 0.125% chitosan solution into sample P0 and shaking the mixture for another 24 h, and it was confirmed that this modification could significantly impact the drug loading efficiency, which reached 94%. In the drug release experiments, the release profiles of P0 and P1 exhibited three stages of release. The initial burst release occurred from 0 to 6 h, followed by a reabsorption stage from 7 to 30 h, and finally a stable sustained release stage was reached after 31 h. As for P2, burst release was also observed in the first 10 h of the experiment. The release curve resembled that of PTX-Chi (C1), indicating that the initial drug release from P2 involved PTX-Chi on the surface of HAp-Gel releasing the drug into the solution. After 25 weeks, the cumulative release percentages were 94.74% for P0, 69.14% for P1, and 33.81% for P2. In the cytotoxicity test, G292 cells were co-cultured with P0, P1, and P2 for 1, 4, 7, and 14 days. After two weeks of cultivation, the cell viability of P0 and P1 decreased to below 20%, while the cell viability of P2 was 56%. Due to the lower drug release amount compared to P0 and P1, P2 did not exhibit strong inhibitory effects over the two-week period, but it revealed a continuing trend of decreasing cell viability over time, indicating that neither loading method affected the efficacy of PTX. The direct addition of chitosan to PTX loaded on HAp-Gel did not affect the drug loading. Furthermore, shifting the order of chitosan addition increased the drug entrapment efficiency. Finally, the addition of chitosan and the shielding effect of the cone-like pore of HAp-Gel microspheres, coupled with the hydrophobic nature of PTX itself, prolonged its drug release duration by more than six months.

## Figures and Tables

**Figure 1 pharmaceutics-15-02025-f001:**
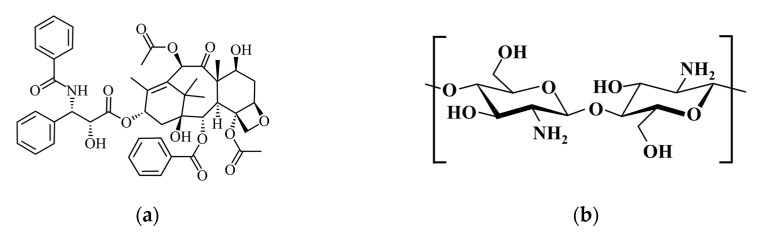
Chemical structures of (**a**) PTX, (**b**) chitosan and (**c**) HAp.

**Figure 2 pharmaceutics-15-02025-f002:**
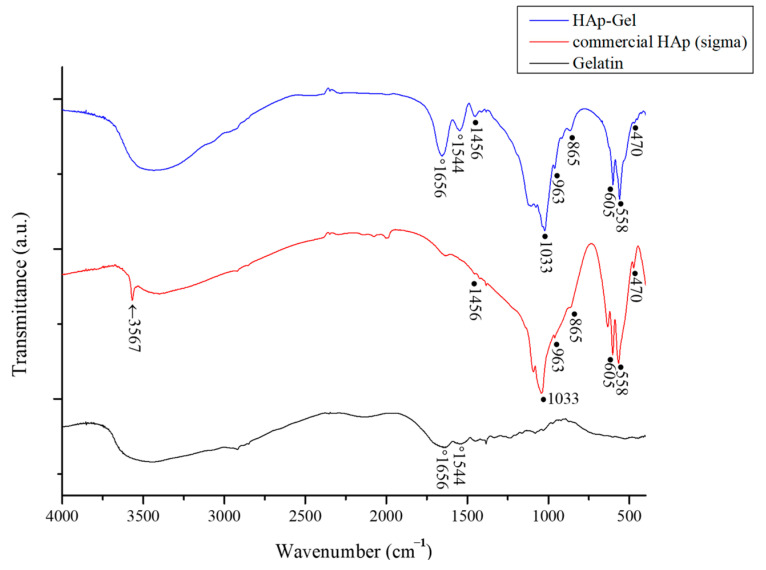
FTIR spectra acquired from HAp-Gel composite microspheres, commercial HAp, and gelatin, respectively.

**Figure 4 pharmaceutics-15-02025-f004:**
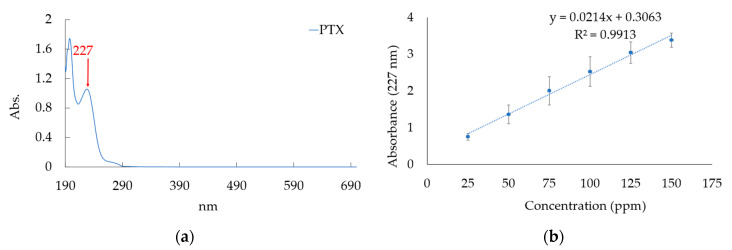
(**a**) The UV-visible spectrum of PTX solution and (**b**) the calibration curve for amounts of PTX dissolved in ethanol solutions ranging from 25 ppm to 150 ppm.

**Figure 9 pharmaceutics-15-02025-f009:**
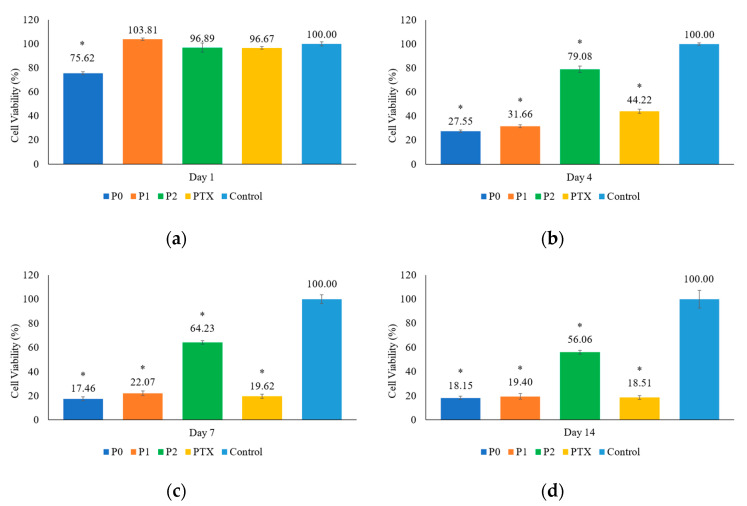
Bar charts of cell viability of G292 cell on days (**a**) 1, (**b**) 4, (**c**) 7, and (**d**) 14 for samples P0, P1, P2, PTX-only, and the control group. * Statistically significant difference between the control group and P0, P1, P2, and PTX groups at *p* < 0.05.

## Data Availability

Data sharing not applicable.

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
