# Peer review of "Effects of Adding Chitosan on Drug Entrapment Efficiency and Release Duration for Paclitaxel-Loaded Hydroxyapatite—Gelatin Composite Microspheres"

_pharmaceutics, 2023, doi:10.3390/pharmaceutics15082025_

Round 1

Reviewer 1 Report

Needs minor review.

Reviewer 2 Report

In order to prepare a system containing hydroxyapatite-gelatin for sustained placlitaxel release, and evaulate the effect of the chitosan-adding processes, the authors offer a straightforward procedure. Fits well with the Aims and Scope of Pharmaceutics. Novelty is moderate.

The introduction makes it clear how important the study is and explains what this investigation's goals are. The proposed methodologies succeed in producing valuable information.

In my opinion, the manuscript is well-written and displays results that currently hinder its publication in its current form. However, I suggest smaller changes for its publication:

In introduction section,

Lines 118-119  “Microtubules also play an essential role in maintaining cell structure, motility, and intracelular movement.” And 121-122,  “The assembly of microtubules takes place during the G2 and pre-phase of mitosis.”

Fits better in the previous paragraph, than cover all of the aspects of microtube properties. This change could facilitate easier reading of both paragraphs.

 In the section on materials and procedures.

Why weren't the samples covered with conductive material in the Surface Morphology section? how results are impacted by this approach change?.

Also in materials and methods, in drug release PTX-HAp-Gel section, is specified that the sample and solution were subjected to centrifugation, in special cases, Why?, what function it has?, and how is this variation to the methodology influences the results?.

The morphology of other prepared materials might have been covered in the section on the gel's surface morphology.

Finally, in Section 3.22 the information contained in Table 1 is repeated, but the Table is not mentioned in the text.

Reviewer 3 Report

In the present manuscript, the authors report the preparation of hydroxyapatite-gelatin microspheres with conical pores were synthesized by a wet chemical method using (NH4)H2PO4 and Ca(NO3)2·4H2O as a source of calcium and phosphate ions with the addition of gelatin, proving to be more osteoconductive than commercial products, and discuss the systems' ability to trap drugs. The work is very interesting, but the biomaterial is not sufficiently characterized, and some suggestions should be considered before its publication in Pharmaceutics.

Below are my suggestions: 

1.      Remove dot from title

2.      In the abstract write the name of the formulae and put the chemical formula in brackets.

3.      The introduction should be strengthened. The authors should compare their chitosan-based biomaterial with others in the literature to emphasise the innovation of their work. Recent papers such as Materials 2022, 15(22), 8208; https://doi.org/10.3390/ma15228208; Int. J. Mol. Sci. 2022, 23(12), 6574; https://doi.org/10.3390/ijms23126574, should be discussed; indeed, I observed that the authors cite papers from more than 30 years ago, when there are several papers on the subject in the current literature.

4.      When the authors discuss gel preparation, they need to add more detail. For example, what molar concentration of the two inorganic salts was used to form the hydroxyapatite?

5.      Line 160: “in a water bath at 65°C”, perhaps you mean at reflux?

6.      No characterization of the new material? E.g. TGA, FTIR and rheological analysis? The authors should provide these further characterizations.

7.      SEM images should also be supported by EDX analysis.

8.      Statistical analysis should be provided for all quantitative data.

9.      Figure 6 c: use the same font size.

10.  There are several typos and grammatical errors; pay attention.

Moderate editing of English language required

Reviewer 4 Report

Comments:

1. The concise Introduction should be conducted with meanwhile more literature review on previous published article for role of chitosan on drug entrapment efficiency and release of hydroxyapatite-gelatin composite should be greatly addressed.

2. "The solutions are then combined, and the HAp-Gel is allowed to precipitate under strictly controlled conditions." Its detail should be revealed.

3. Materials and all chemicals of test should be informed for their sources including batch/lot, brand, company, city/state, country.; especially for chitosan should be revealed for its M.W., %DA and source of production: squid pen, crab, shrimp or any sources. For gelatin type, its crucial detail is needed also for stated. Same as to apply for all related instruments. Cell source and its passage data are needed to inform.

4.  The condition for SEM experiment and drug analysis via UV-Visible spectroscopy should be revealed and how the residue or will other dissolved compound interfere drug absorbance and also at 2.4 for drug release with direct contact method?

5. Why was 0.0033 vol% acetic acid employed for dissolving chitosan please provide the reason for this used concentration and how is it affected on gelatin and composite such as their solubility? Additionally, why on single concentration of chitosan 0% or 0.125% is used not varied for optimum one in 2.3.2 and 2.3.3.

6. pH value of "phosphate-buffered saline (PBS) 215 solution: in Line 215 should be included and why to use this type of this buffer.

7. Besides sem, the other characterization techniques such as FTIR, dsc and others should be performed to confirm the results. SEM after release test should be included.

8. S.D. should be include in Fig.2 and Table 1. and others both in experiment and results.

9. More discussion with appropriate supporting refs is needed in 3.2.2, 3.3, 

10. Reabsorption of drug phenomena during drug release should be supported with evidence-based or supporting refs. And please explain for uncomplete drug liberation and its disadvantage or how to solve.

11. The suggestion for applying in real situation of cancer treatment of this developed microsphere should be included and further on-going tests. 

12. Reference lists are not prepared consistently.

The grammar should be corrected through the manuscript.

Round 2

Reviewer 1 Report

None

Reviewer 3 Report

Suitable corrections were made. I accept this manunuscript in present form.